# Evaluation of Fungal Volatile Organic Compounds for Control the Plant Parasitic Nematode *Meloidogyne incognita*

**DOI:** 10.3390/plants12101935

**Published:** 2023-05-09

**Authors:** Pasqua Veronico, Nicola Sasanelli, Alberto Troccoli, Arben Myrta, Audun Midthassel, Tariq Butt

**Affiliations:** 1Institute for Sustainable Plant Protection, CNR, Via G. Amendola 122/D, 70126 Bari, Italy; nicola.sasanelli@ipsp.cnr.it (N.S.); alberto.troccoli@ipsp.cnr.it (A.T.); 2Certis Belchim BV, Stadsplateau 16, 3521 AZ Utrecht, The Netherlands; arben.myrta@certisbelchim.com (A.M.); audun.midthassel@certisbelchim.com (A.M.); 3Department of Biosciences, Swansea University, Singleton Park, Swansea SA2 8PP, UK; t.butt@swansea.ac.uk

**Keywords:** root-knot nematodes, *Meloidogyne incognita*, VOCs, 1-Octen-3-ol, 3-Octanone

## Abstract

Plant parasitic nematodes are a serious threat to crop production worldwide and their control is extremely challenging. Fungal volatile organic compounds (VOCs) provide an ecofriendly alternative to synthetic nematicides, many of which have been withdrawn due to the risks they pose to humans and the environment. This study investigated the biocidal properties of two fungal VOCs, 1-Octen-3-ol and 3-Octanone, against the widespread root-knot nematode *Meloidogyne incognita*. Both VOCs proved to be highly toxic to the infective second-stage juveniles (J2) and inhibited hatching. Toxicity was dependent on the dose and period of exposure. The LD50 of 1-Octen-3-ol and 3-Octanone was 3.2 and 4.6 µL, respectively. The LT50 of 1-Octen-3-ol and 3-Octanone was 71.2 and 147.1 min, respectively. Both VOCs were highly toxic but 1-Octen-3-ol was more effective than 3-Octanone. Exposure of *M. incognita* egg-masses for 48 h at two doses (0.8 and 3.2 µL) of these VOCs showed that 1-Octen-3-ol had significantly greater nematicidal activity (100%) than 3-Octanone (14.7%) and the nematicide metham sodium (6.1%). High levels of reactive oxygen species detected in J2 exposed to 1-Octen-3-ol and 3-Octanone suggest oxidative stress was one factor contributing to mortality and needs to be investigated further.

## 1. Introduction

Plant parasitic nematodes (PPNs) cause severe damage and significant yield losses of a wide range of important crops with the estimated worldwide annual losses of US $157 billion/year [1]. Root knot nematodes (RKNs; *Meloidogyne* spp.) rank among the most destructive PPN species infecting the roots of almost all cultivated plant species. The nematode population density at transplant or sowing can significantly influence plant growth [2]. Symptoms of nematode attack (yellowing, stunting growth, wilting, dwarfism) can be aggravated by the presence of fungal and bacterial pathogens which utilize PPN penetration pathways to gain entry into the plant and cause disease [3,4,5,6]. The severity of PPN damage is expected to increase due to global warming and intensification of agriculture to meet the needs of a rapidly growing world population [7]. Current control of PPN is still dependent on the use of synthetic nematicides but many chemicals have been withdrawn or restricted in their use due to the risks they pose to human health, pollution of the environment and other drivers, including the European Green Deal [8]. More attention is being given to sustainable control tools, including pesticides of natural origin or “biopesticides” [9]. A wide range of agronomic strategies are being used, including soil amendments, green manures, bio-fumigation, crop rotations, cover crops, grafting, resistant cultivars, soil solarization, and steam sterilization [10,11,12,13,14,15,16,17]. There is increasing interest in biopesticides, including botanicals or plant derived essential oils [18,19,20,21] and fungal and bacterial biocontrol agents [22,23,24].

In this context, the use of volatile organic compounds (VOCs) of natural origin (plants/microbes) against plant pathogens and parasites provides an interesting and promising ecofriendly alternative to the use of chemical pesticides [25]. They are characterized by a high vapor pressure, evaporate at room temperature, and a low water solubility. Several VOCs have shown promise in the management of invertebrate plant pests [26,27,28] including PPNs [29,30,31,32]. VOCs vary in chemistry and biological properties and in some cases exhibiting greater nematicidal activity than commercial pesticides [27]. VOCs of entomopathogenic fungi belonging to the genera *Metarhizium* and *Beauveria* influence insect behaviour [33,34]. Recently, two VOCs, 1-Octen-3-ol and 3-Octanone which are produced by both plants and fungi, [35,36,37] were shown to have biopesticide properties. Depending on the dose, these compounds repelled or killed molluscs [38], influenced *Penicilliun paneum* spore germination [39], attracted or repelled insects [40,41], induced a defensive response in *Arabidopsis thaliana* [42] and influenced the behaviour of the northern root-knot nematode *Meloidogyne hapla* on susceptible plants [27]. The potency and ephemeral nature of 1-Octen-3-ol and 3-Octanone make them promising candidates for development as new bio-nematicides.

This study investigated the in vitro toxicity of these two VOCs on juveniles (J2) and egg masses of the most widespread and damaging RKN, *M. incognita*, providing new insights as to the mode of action of these nematicidal volatiles.

## 2. Results

### 2.1. Nematicidal Activity on M. incognita Juveniles

Both 1-Octen-3-ol and 3-Octanone were toxic to *M. incognita* with mortality being dose and time dependent (Table 1 and Table 2).

The lowest doses of 1-Octen-3-ol caused over 50% mortality from 3 h post treatment. Highest doses were more effective, causing 50% mortality from 1.5 h. All doses caused about 100% mortality at the highest exposure time (Table 1).

At 24 h, 3-octanone caused 100% mortality only at the highest doses. Mortality of more than 90% was evident from 3 h post treatment. At the lowest dose (2.5 µL), 38.3% juveniles were killed after 6 h and mortality never reached 100% even after 24 h post treatment (Table 2). Probit analysis was used on the base of the corrected mortality to calculate values of LD_50_ and LT_50_ at each dose and exposure time, respectively (Table 3).

Values of LD_50_ ranged between 10.1 and 0.2 µL of 1-Octen-3-ol for the exposure times between 45 min and 12 h. Values of LT_50_ ranged between 91.6 and 40.1 min (Table 3).

The mean LD_50_ value of 3-Octanone was higher than that of 1-Octen-3-ol. LD_50_, ranging between 8.5 and 0.6 µL for the different considered exposure times. LT_50_ values were also higher than those of 1-Octen-3-ol, ranging between 399.3 and 39.6 min for the different considered doses (Table 3). The average LD_50_ and LT_50_ values for 1-Octen-3-ol were 3.2 µL and 71.2 min, respectively. These values were lower than the average LD_50_ (4.6 µL) and LT_50_ (147 min) values for 3-Octanone showing 1-Octen-3-ol to be more efficacious as a nematicide than 3-Octanone (Table 3). For both VOCs, high and significant negative correlation indices (power equation *y = bx^m^*) were found between: (a) LD_50_ and exposure times and (b) LT_50_ and doses (Table 3).

### 2.2. Nematicidal Activity of VOCs on M. incognita Egg Masses

All treatments were significantly effective in reducing egg viability. 1-Octen-3-ol caused 100% mortality at both doses (0.8 and 3.2 µL). There was zero mortality in the untreated controls (Figure 1). The VOC 3-Octanone was less effective than 1-Octen-3-ol and no statistical difference was observed between the two applied rates. 3-Octanone used at 0.8 and 3.2 µL caused 8 and 14.7% mortality, respectively (Figure 1). The chemical control metham sodium, used at the rate 0.1 µL/1 mL (*v*/*v*), corresponding to minimum recommended dose for field application (400 L/Ha), was the least effective in reducing vitality of eggs inside the gelatinous matrix of *M. incognita* egg masses (6.1%), although the percentage of mortality was not statistically different from that observed in the 0.8 µL 3-Octanone treatment (Figure 1).

### 2.3. Reactive Oxygen Species Induction in M. incognita Juveniles by 1-Octen-3-ol and 3-Octanone

Reactive oxygen species (ROS) production was elevated in J2 exposed to 1-Octen-3-ol and 3-Octanone. Nematodes died following exposure to the highest dose (20 µL) of both VOCs. They were considered dead as they did not move after being rinsed 3 times in sterile distilled water and stimulated with a needle. Morphological changes were clearly detectable after 12 and 24 h of treatment with 1-Octen-3-ol as fumigant. Pharyngeal and intestinal tissues became indistinct and large empty vacuoles were visible, extending throughout the intestine (Figure 2F,H,J,L). No disruption was observed in the intestine or pharyngeal tissues of the control group exposed only to water (Figure 2A,C,E,G,I,K). A time course (6, 12, 24 h) study of oxidative stress in nematodes exposed to the two VOCs showed that 1-Octen-3-ol elevated ROS several hours before that of 3-Octanone (Figure 2, Figure 3 and Figure 4). Little or no autofluorescence was detected in nematodes excited with light at 450–490 nm wavelength irrespective of being controls or exposed to 1-Octen-3-ol (Figure 2A’,B’,E’,F’,I’,J’). Fluorescence was detected in nematodes treated with dichlorofluorescein diacetate (DCFH-DA). Small autofluorescent spots were detected in the intestine of live control nematodes (Figure 2C’,G’,K’) but significant fluorescence due to the generation of ROS was observed in the head and pharynx of J2 exposed for 12 h to 1-Octen-3-ol (Figure 2H’). The fluorescence intensity increased extended along the body length of nematodes treated for 24 h (Figure 2L’).

Nematodes exposed to 1-Octen-3-ol exhibited severe oxidative stress at 12 and 24 h since the intensity of fluorescence was respectively 13 and 10.2-fold higher in treated nematodes versus untreated controls (Figure 3).

Disruption of the intestine and pharynx of *M. incognita* J2 was also observed when nematodes were treated with 3-Octanone (Figure 4F,H,J,L). Autofluorescence was initially similar for untreated and treated nematodes (Figure 4A’,B’,E’,F’,I’,J’). However, fluorescence, indicative of ROS production, was detectable 24 h post treatment (Figure 4L’) with its intensity being 14-fold higher than in the control (Figure 5).

## 3. Discussion

The VOCs 1-Octen-3-ol and 3-Octanone were shown to be highly toxic to the infective J2 stage of the southern root-knot nematode *M. incognita*, while 1-Octen-3-ol was also highly toxic to egg masses, significantly reducing egg viability. The fact that ROS detected using fluorescent probes were elevated earlier in J2 exposed to 1-Octen-3-ol than 3-Octanone confirmed that it was faster acting and more potent than 3-Octanone. Since both VOCs kill J2 stage of *M. incognita* and *M. hapla* [27] further demonstrates their potential for development as future nematicides. The current study corroborates the findings of Khoja et al. [27] that mortality is dependent on the dose and time post exposure. Thus, high doses kill quickly while low doses take longer to kill. Clearly, these volatiles are causing irreversible damage, ultimately leading to death. Killing or debilitating the J2 stage is vitally important as this is the stage that infects plants. Preventing infection of host plants reduces crop damage and infection by opportunistic pathogens. Many rhizosphere fungi, including EPF like *Metarhizium*, produce 1-Octen-3-ol and 3-Octanone but it is unclear if these volatiles reduce field populations of PPN or protect plants from infections. Several bacterial species have been shown to produce an array of nematicidal VOCs which are toxic to RKN in vitro and in a crop [31,32]. Since the profile varies with the bacterial species suggest that each species evolved nematicidal compounds independently [43,44,45].

The current study shows that, at the applied doses, 1-Octen-3-ol is superior to 3-Octanone and metham sodium in reducing egg viability. Fumigants with ovicidal properties are a desired feature of potential commercial nematicides as eggs represent the main survival source of the *M. incognita* population in the soil [46]. Interestingly, egg mortality recorded at the lowest dose of 1-Octen-3-ol was about 9-fold higher than that observed when metham sodium was applied at the minimum recommended dose. It has been reported that many VOCs from fungi, *Fusarium oxysporum* [47] and *Duddingtonia flagrans* [48], and bacteria *Pseudomonas putida* and *Virgibacillus dokdonensis* [30,49] affect egg hatching, but this activity has not been previously described for 1-Octen-3-ol and 3-Octanone. We can hypothesize that VOCs are sequestered in the gelatinous matrix where eggs are embedded creating a toxic environment for the eggs and emergent J2. The significant difference in egg mortality between 1-Octen-3-ol and 3-Octanone could be attributed to differences in their ability to penetrate through eggshell, but this hypothesis requires further investigation.

To promote a deleterious effect on a pest, the VOCs need to penetrate the tissues to cause damage [38]. In particular, in insects and molluscs their hard cuticle and shell act as a stronger barrier which limit/block diffusion of the VOCs [28]. However, these compounds appear to easily cross the RKN cuticle. This assumption is based on the damage caused by1-Octen-3-ol and 3-Octanone to the internal intestinal tissue which was less defined, suggesting lysis or cell disruption similarly to what observed in previous studies with other nematicidal VOCs [49,50]. 1-Octen-3-ol exhibits strong antimicrobial activity, eliciting autolysis especially in yeast and bacteria [51]. The current study shows that exposure of *M. incognita* J2 to 1-Octen-3-ol and 3-Octanone as fumigants in a closed environment resulted in elevated levels of ROS, and caused severe oxidative stress, inducing damage to internal structures and death. The fluorescence observed in ROS detection assay upon treatment with both VOCs could be attributed to hydrolysis of dichlorofluorescein diacetate by esterases and proteases released in damaged tissues and therefore leading to the entire body of nematode to fluoresce. Our results match previous studies which reported that exposure of *Drosophila melanogaster* to 1-Octen-3-ol induced oxidative stress by increasing ROS levels accompanied by stimulation of glutathione-S-transferase (GST) and superoxide dismutase (SOD) [52]. A similar mode of action was also reported for the VOCs dimethyl disulfide (DMDS), methyl isovalerate (MIV), and 2-undecanone (2-UD) produced by the GBSC56 strain of *Bacillus* sp. which showed strong nematicidal activity on *M. incognita* as well [53]. The ROS detection assay highlighted that the damage is spread with time. It could be possible that in the first hours after exposure nematodes try to mitigate the damage by scavenging ROS but the damage caused means the nematode is under increasing stress. ROS begin to accumulate in the anterior part of the body and then disperse along the body as exposure time increases. Accumulation of ROS is delayed in J2 exposed to 3-Octanone which correlates with it being less toxic than 1-Octen-3-ol where ROS was observed at least 12 h earlier. Altogether, this study has shown that 1-Octen-3-ol and 3-Octanone show much promise as nematicides with the former being more toxic to J2 but also eggs. Mortality can be partly attributed to stress and internal damage to body tissues. However, further work is needed to determine the efficacy of the VOCs as plant protection products in a growing crop.

## 4. Materials and Methods

### 4.1. Reproduction of the Plant Parasitic Nematodes Meloidogyne incognita Race 2

*Meloidogyne incognita* (Kofoid and White) Chitw, race 2 was used in all experiments. Pure populations of *M. incognita* race 2 were maintained on tomato plants (*Solanum lycopersicum* cv Rutgers) in a greenhouse at 25 ± 2 °C. The species was previously identified by morphological and molecular patterns and classified as race 2 with differential host tests [54]. Eggs were extracted from infested roots and incubated in hatching chambers at 26 ± 2 °C. Only fresh hatched J2 were used in the experiments. 

### 4.2. Volatile Organic Compounds (VOCs)

The two VOCs 1-Octen-3-ol (CAS number 3391-86-4) and 3-Octanone (CAS number 106–68-3) purchased from Sigma-Aldrich (≥98% purity), have a concentration of 784 g/L. 1-Octen-3-ol, known as mushroom alcohol, is a secondary alcohol derived from 1-Octene. It is approved by the U.S. Food and Drug Administration as a food additive. It has 128.21 molecular weight; insoluble in water, soluble in oils and miscible at room temperature in ethanol, solubility of 1.836 mg/mL at 25 °C (est). Vapor pressure is 3 hPa at 20 °C. The LD50 doses are 56, 340 and 3300 mg/Kg in mouse, rat and rabbit, respectively [55].

3-Octanone, also known as ethyl amyl ketone appears as a clear colorless liquid with a pungent odor. Partially soluble in alcohol and insoluble in water, solubility of 2.6 mg/mL. Vapors are denser than air and may have a narcotic effect in high concentrations. It is used in perfumes and as a solvent for nitrocellulose and vinyl resins. It has the same molecular weight and formula of 1-Octen-3-ol (C8H16O). Vapor pressure is 2.7 hPa at 20 °C. The LD50 is 0.406 and 5 g/Kg in mouse and rat, respectively [56]. 

### 4.3. In Vitro Tests

#### 4.3.1. Evaluation of VOCs for Their Nematicidal Activity on *M. incognita* Juveniles

*Meloidogyne incognita* J2 were used to carry out in vitro tests in which the effects of 1-Octen-3-ol and 3-Octanone were assessed on the mortality of the nematode using different doses (0, 2.5, 5.0, 10.0 and 20.0 µL) and exposure times (45 min, 1.5, 3, 6, 12 and 24 h). A single control (0 dose) was used for both VOCs. There were 5 replications for each combination exposure time x VOC dose.

The experiment aimed to determine the median lethal dose (LD50) and median lethal time (LT50) of the two VOCs. Median lethal dose and median lethal time are the amount of VOC able to cause the death of 50% of a group of test nematode (acute toxicity) and the average time during 50% of a population may be expected to die following acute administration of the chemical at a given dose, respectively.

A Petri dish assay with 3.8% *w*/*v* water agar (Sigma Aldrich, Saint Louis, MO, USA) was set up to determine the nematicidal effect of the two VOCs according to the procedure described by Khoja et al. [27] with some modifications. Six circles (1 cm diameter) for nematode sampling zone (Figure 6A) were drawn on each 9 cm diameter Petri dish.

Twenty microliters µL of a water *M. incognita* J2 suspension containing about 10–15 specimens were pipetted in each circle. Nematodes were exposed to different doses (2.5, 5, 10 and 20 μL) of the tested VOCs released from 0.5 cm diameter Whatman filter paper attached to a glass coverslip (22 mm × 22 mm) put on the centre of the Petri dish lid (Figure 6A). Petri dishes were closed and firmly sealed with parafilm to avoid the escape of volatiles and incubated at room temperature in the dark. The same Petri dish, for each applied dose, was observed for the survival and motility of the nematode at 45 min, 1.5, 3, 6, 12 and 24 h.

Averages of the number of J2 observed in plates treated with 1-Octen-3-ol and 3-Octanone, independently from the sampling zone, were 54 and 78, respectively.

The survival of the nematodes was distinct for each sampling area (centre, middle and external sectors). At each exposure time the number of living and dead J2 was observed and the relative percentage was calculated. The dead and alive nematodes in these circles were counted using a stereo microscope at 20× magnification (Nikon SMZ745T). Nematodes were classified as dead if they were unable to recover their motility after being stimulated with a needle in additional Petri dishes observed at the same dose and exposure time. As the tested products are volatile compounds and they spread uniformly in the space of the Petri dish the average of five replications was calculated for the different exposure times and doses. The study was performed twice.

The percentages of mortality were corrected by eliminating the natural death in the control according to Schneider-Orelli’s formula [57]: Corrected mortality% = [(Mortality% in treatment − Mortality% in the control) (100 − Mortality% control)] × 100.

#### 4.3.2. Evaluation of VOCs for Their Nematicidal Activity on *M. incognita* Egg Masses

The experiment was limited for both VOCs to 5 and 20 µL doses used in the previous test on J2 in Petri dishes (63 mL volume). These treatments were compared to metham sodium at 0.1 µL/mL (*v*/*v*) (Taminco Italia s.r.l.) (chemical control) and to an untreated control considering four replications per treatment.

Twenty-five similar size egg masses were hand-picked and put in a small (2 mL) plastic tube for the different treatments. The plastic tubes were then introduced in a polyethylene tube of 10 mL volume (Figure 6B1). A filter paper was inserted under the cap of the tube to allow treatments (Figure 6B2).

The amounts of each VOC applied on the filter paper (0.8 and 3.2 µL), which corresponded to doses of 5 or 20 µL, were calculated according to 10 mL volume of the tube. Tubes immediately after the treatment were closed and sealed with parafilm (Figure 6B3). Metham sodium was used at the rate of 1 µL/10 mL volume, according to the minimum dose allowed in field treatments [17]. Egg masses after two days of exposure were removed from the tubes and placed on 2 cm diameter sieves (215 µm aperture) (Figure 6B4). Each sieve was put in a 3.5 cm diameter Petri dish. Dishes were incubated in a growth cabinet at 25 ± 2 °C. Emerged J2 were counted at weekly intervals, renewing filtered tap water at the same time, throughout 9 weeks (6B5–6). At the end of the hatching test, egg masses were shaken for 3 min in a 1% sodium hypochlorite aqueous solution [58] and the unhatched eggs were counted. Numbers of J2 emerging weekly were expressed as cumulative percent of the total initial population (hatched + unhatched eggs).

#### 4.3.3. ROS Production

The mechanism of action of 1-Octen-3-ol and 3-Octanone were evaluated setting up a ROS production assay to ascertain if the volatile treatment activated oxidative damage. One hundred *M. incognita* J2 per plate were exposed to 20 µL of 1-Octen-3-ol or 3-Octanone in agar plates (3.8%) by incubation for 6, 12 and 24 h at 25 °C in the dark. In the control plate nematodes were treated with sterile distilled water (0 dose) for the same time as for treatments. Juveniles after exposure to VOCs were collected in small beakers by rinsing the plate with 1 mL of sterile distilled water, then transferred in 1.5 mL Eppendorf tubes and centrifuged for 1 min at 12,000 rpm at room temperature. The collected nematodes were then treated for 30 min at 25 °C with a mixture of M9 buffer pH 7 and 10 µM 2′,7′-dichlorofluorescein diacetate (DCFH-DA) (Sigma-Aldrich, Saint Louis, MO, USA). Samples were observed with a Leica DM 4500 B light microscope (Leica Microsystems, Milan, Italy) using an I3 filter cube (excitation filter BP 450–490 nm and detection filter LP 515 nm). The images were recorded by a Leica DFC 450C camera. Fluorescence intensity from fluorescence microscopy images was determined by ImageJ software (National Institute of Health, Bethesda, MA, USA). The corrected fluorescence (CTF) was calculated by applying the following formula CTF = Integrated Density − Area of selected nematode × Mean fluorescence of background readings). Five specimens were examined per treatment.

### 4.4. Statistical Analysis

Data from the experiment on J2 were subjected to analysis of variance (ANOVA) and means compared by the Least Significant Difference’s Test (*p =* 0.05 and *p =* 0.01). For both VOCs data of percent corrected mortality were subjected to probit analysis [59] to estimate values of lethal doses (LD50) and lethal times (LT50). The highest negative correlations between applied doses and exposure times were obtained using the power interpolation formula: y = b × m.

Percentage data from the hatching test derived from egg masses, treated for 24 h with 20 µL of both VOCs, were arcsin square root transformed [60] and subjected to analysis of variance (ANOVA), comparing means by Least Significant Difference’s Test. For both experiments data on nematode mortality rates were calculated on the base of results (dead or alive J2 and J2 hatched) and corrected according to Schneider-Orelli’s formula to eliminate the natural death in the controls. Statistical analysis was performed using the software Plot IT Version 3.2 (Scientific Programming Enterprises, Haslett, MI, USA). Fluorescence intensity, between treatments and control, was analysed by Student’s *t*-test (*p* < 0.05).

## Figures and Tables

**Figure 1 plants-12-01935-f001:**
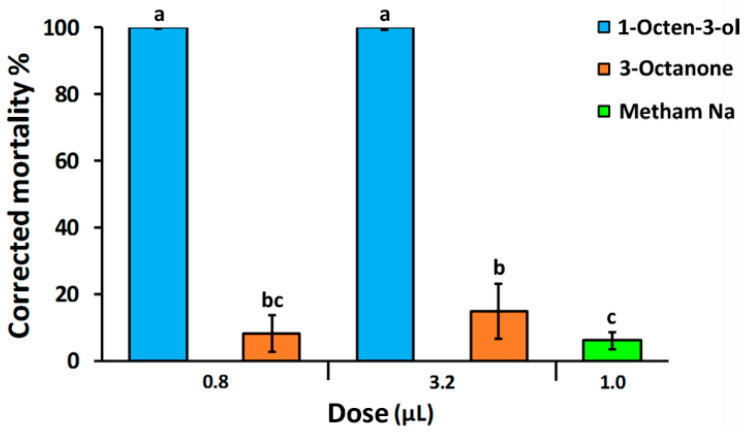
Changes in corrected mortality of *Meloidogyne incognita* eggs after exposure to 1-Octen-3-ol and 3-Octanone at different doses. Values were calculated relative to untreated control. Metham sodium was applied at the minimum recommended dose for field application. Bars with the same letters are not statistically different according to the Least Significant Difference’s test at *p =* 0.05.

**Figure 2 plants-12-01935-f002:**
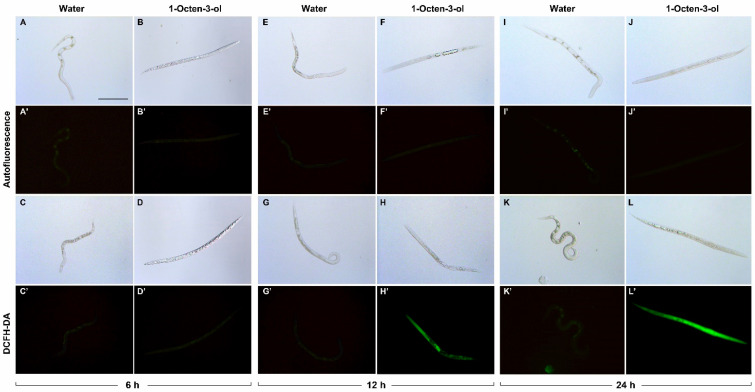
ROS production in *Meloidogyne incognita* juveniles exposed to 1-Octen-3-ol. The same specimen was photographed under light and fluorescence sources. Comparison of intestinal autofluorescence in nematodes exposed to sterile distilled water (controls) or 20 µL of 1-Octen-3-ol for 6 h (**A**,**A’**,**B**,**B’**), 12 h (**E**,**E’**,**F**,**F’**) and 24 h (**I**,**I’**,**J**,**J’)**. Comparison of ROS induction in controls (**C**,**C’**,**G**,**G’**,**K**,**K’**) and 1-Octen-3-ol treated juveniles (**D**,**D’**,**H**,**H’**,**L**,**L’**) during the time course. Scale bar = 100 µm.

**Figure 3 plants-12-01935-f003:**
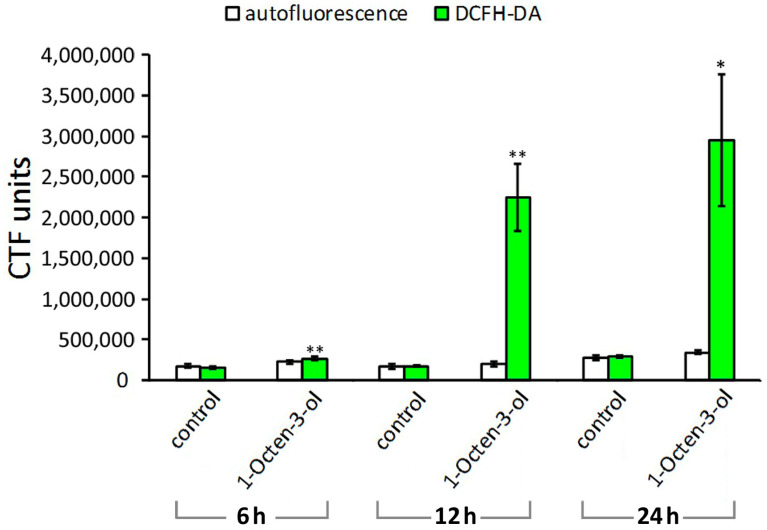
Comparison of autofluorescence and ROS production in nematodes exposed to 1-Octen-3-ol versus control nematodes exposed to distilled H_2_O. The bars represent mean ± SE. Significant differences between treated and untreated juveniles according to Student’s *t*-test at **p* < 0.05; ***p* < 0.01.

**Figure 4 plants-12-01935-f004:**
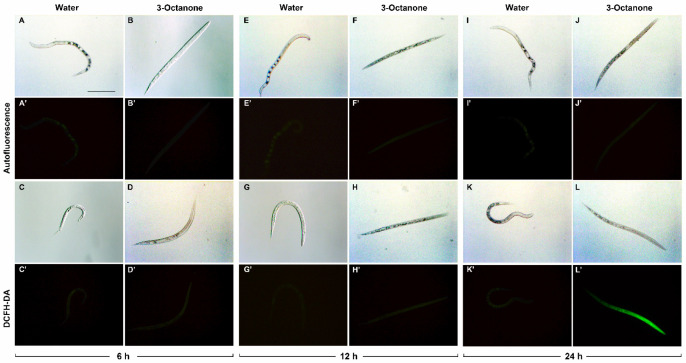
ROS production in *Meloidogyne incognita* juveniles exposed to 3-Octanone. The same specimen was photographed under light and fluorescence sources. Comparison of intestinal autofluorescence in nematodes exposed to sterile distilled water (controls) or 20 µL of 3-Octanone for 6 h (**A**,**A’**,**B**,**B’**), 12 h (**E**,**E’**,**F**,**F’**) and 24 h (**I**,**I’**,**J**,**J’**). Comparison of ROS induction in controls (**C**,**C’**,**G**,**G’**,**K**,**K’**) and 3-Octanone treated juveniles (**D**,**D’**,**H**,**H’**,**L**,**L’**) during the time course. Scale bar = 100 µm.

**Figure 5 plants-12-01935-f005:**
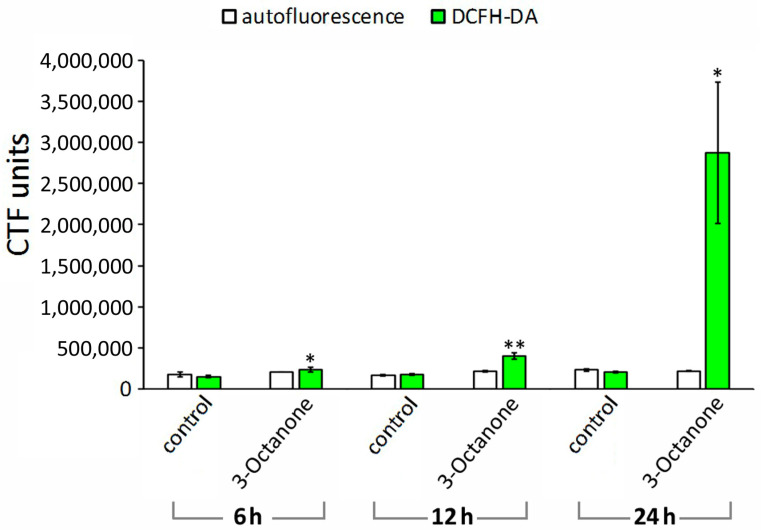
Comparison of autofluorescence and ROS production in nematodes exposed to 3-Octanone versus control nematodes exposed to distilled H_2_O. The bars represent mean ± SE. Significant differences between treated and untreated juveniles according to Student’s *t*-test at * *p* < 0.05; ** *p* < 0.01.

**Figure 6 plants-12-01935-f006:**
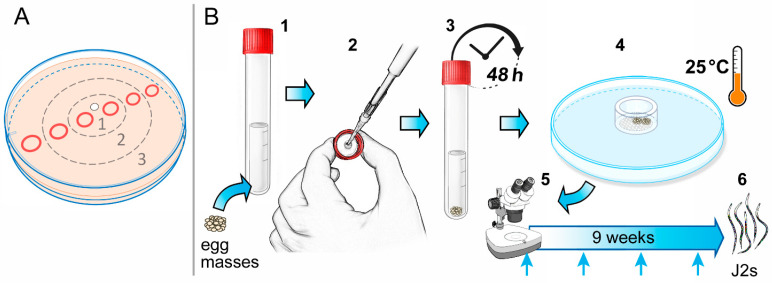
Procedures followed to assess the nematicidal effect of 1-Octen-3-ol and 3-Octanone on *M. incognita* J2 and egg masses. (**A**) Juveniles were loaded on a plastic Petri dish divided into sectors (1 centre, 2 middle, 3 external); (**B1**) Egg masses were put in tubes inserted in larger capped tubes; (**B2**–**B3**) VOCs were applied on filter paper inserted under the cap and tubes were closed and sealed with parafilm; (**B4**–**B6**) Batches of treated egg masses to use in the hatching assay.

**Table 1 plants-12-01935-t001:** Corrected per cent mortality of *Meloidogyne incognita* J2 exposed to 1-Octen-3-ol.

Exposure Time		Doses (µL)				LSD	
		2.5	5.0	10.0	20.0	0.05	0.01
45 min		38.0 * ± 9.8 **	40.0 ± 11.8	46.6 ± 16.2	60.0 ± 8.9	22.6	33.0
1.5 h		46.7 ± 18.9	48.7 ± 20.5	50.0 ± 20.5	63.7 ± 17.0	36.3	52.8
3 h		65.7 ± 15.5	68.0 ± 12.6	74.6 ± 20.5	77.7 ± 12.3	29.8	42.7
6 h		74.3 ± 6.5	69.3 ± 16.8	82.3 ± 3.2	82.7 ± 3.2	17.5	25.5
12 h		89.3 ± 3.2	96.0 ± 3.5	97.3 ± 1.2	99.3 ± 1.2	4.7	6.8
24 h		99.7 ± 0.6	100.0 ± 0.0	99.3 ± 1.2	100.0 ± 0.0	1.2	1.8
LSD	0.05	19.9	23.1	24.3	16.8	---	---
	0.01	27.8	32.4	34.1	23.5	---	---

* Each value is the average of the observations done in the centre, middle and external circles in which Petri dishes were divided (*n* = 5). ** Average ± SD. Least significance differences (LSD) at *p =* 0.05 and 0.01 are provided for exposure time and doses.

**Table 2 plants-12-01935-t002:** Corrected per cent mortality of *Meloidogyne incognita* J2 exposed to 3-Octanone.

Exposure Time		Doses (µL)				LSD	
		2.5	5.0	10.0	20.0	0.05	0.01
45 min		18.0 * ± 9.2 **	52.3 ± 11.7	55.7 ± 10.3	61.0 ± 5.6	17.8	25.9
1.5 h		4.3 ± 2.9	33.0 ± 9.5	60.0 ± 2.0	65.0 ± 3.6	10.1	14.8
3 h		10.3 ± 6.1	51.3 ± 11.0	92.7 ± 1.5	98.7 ± 1.5	12.0	17.5
6 h		38.3 ± 10.8	64.7 ± 10.7	95.0 ± 1.0	99.0 ± 1.2	14.4	20.9
12 h		80.3 ± 3.1	85.3 ± 3.2	97.0 ± 1.0	99.3 ± 1.0	4.4	6.4
24 h		88.0 ± 3.0	92.3 ± 3.2	100.0 ± 0.0	100.0 ± 0.0	4.1	6.0
LSD	0.05	11.8	16.0	7.7	5.1	---	---
	0.01	16.5	22.4	10.8	7.1	---	---

* Each value is the average of the observations done in the centre, middle and external circles in which Petri dishes were divided (*n* = 5). ** Average ± SD. Least significance differences (LSD) at *p =* 0.05 and 0.01 are provided for exposure time and doses.

**Table 3 plants-12-01935-t003:** Median lethal dose (LD_50_) and median lethal time (LT_50_) for the root-knot nematode *Meloidogyne incognita* exposed to different doses of 1-Octen-3-ol and 3-Octanone (0, 2.5, 5, 10 and 20 µL) and different times (45 min, 1.5, 3, 6, 12 and 24 h).

Time	LD_50_ (μL)		Dose (μL)	LT_50_ (min)	
	1-Octen-3-ol	3-Octanone		1-Octen-3-ol	3-Octanone
45 min	10.1 (6–17) *	8.5 (7–11)	2.5	91.6 (73–114)	399.3 (343–464)
1.5 h	5.1 (2–10)	10.0 (8–12)	5.0	86.2 (69–107)	106.5 (80–141)
3 h	0.3 (0.01–6)	4.9 (4–5)	10.0	66.8 (53–85)	43.1 (33–57)
6 h	0.2 (0.005–5)	3.3 (3–4)	20.0	40.1 (28–57)	39.6 (31–51)
12 h	0.2 (0.03–1.5)	0.8 (0.4–1.7)			
24 h	--- **	0.6 (0.2–1.9)			
Average	3.2	4.6		71.2	147.1
Correlations	*y = bx^m^*	*y = bx^m^*		*y = bx^m^*	*y = bx^m^*
	b = 3665	b = 375.7		b = 146.6	b = 842.75
	m = −1.598	m = −0.875		m = −0.394	m = −1.13
	r = −0.92	r = −0.95		r = −0.94	r = −0.94

* In parentheses the fiducial limits; **---Not calculated.

## Data Availability

Data are contained within the article.

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
