# Peer review of "Evaluation of Fungal Volatile Organic Compounds for Control the Plant Parasitic Nematode Meloidogyne incognita"

_plants, 2023, doi:10.3390/plants12101935_

Round 1

Reviewer 1 Report

The nematicidal activity of two volatile organic compounds, 1-Octen-3-ol and 3-Octanone, against juveniles and egg masses of M. incognita was tested in vitro tests. The manuscript is well written and the results clearly presented. Some minor comments need consideration

Consider to express time in hours instead minutes.

L73 latest exposure time or largest (highest) exposure time?

L83 change at the latest exposure time for at 24 h.

L85 at the lowest dose: Indicate 2.5 in parenthesis

L85 ...were killed between 360 and 720 minutes. Rephrase. Only  38.3% were killed at 360 min.

L174 Delete or rephrase “Indeed, more than a commercial chemical nematicide”.  Authors did some in vitro tests but there is not sufficient information in the ms. to support this statement. Metham sodium was tested with egg masses but not with juveniles. The target of a fumigant such as metham sodium, which is always applied before planting, are the juveniles free in the soil and not the egg masses in the roots.

L223 ....animal is under increasing stress.

L 279 In MxM authors indicate that “the survival of the nematodes was distinct for each sampling area (centre, middle and external sectors) ......and average of the five replications was calculated for the different exposure times, doses and sectors. However, only the average of the three sectors is presented in Tables 1 and 2. No information on the distribution of the nematodes per sector is provided in the result section. I assume their distribution per sector was not meaningful probably due to the low numbers of nematodes used as inoculum in each circle. Please clarify and comment in the revised version.

 L295 Volume of the small plastic tube?

Figure 6. Indicate in the legend that 1= centre, 2= middle and 3 external

Author Response

Consider to express time in hours instead minutes.

We expressed all the times of exposure in hours except for 45 minutes

L73 latest exposure time or largest (highest) exposure time?

We changed in highest

L83 change at the latest exposure time for at 24 h.

We changed accordingly

L85 at the lowest dose: Indicate 2.5 in parenthesis

Done

L85 ...were killed between 360 and 720 minutes. Rephrase. Only  38.3% were killed at 360 min.

We changed accordingly

L174 Delete or rephrase “Indeed, more than a commercial chemical nematicide”.  Authors did some in vitro tests but there is not sufficient information in the ms. to support this statement. Metham sodium was tested with egg masses but not with juveniles. The target of a fumigant such as metham sodium, which is always applied before planting, are the juveniles free in the soil and not the egg masses in the roots.

The sentence has been deleted.  The juveniles free in the soil are, indeed, the target of fumigants. However, in field conditions egg masses present in crop residues may represent a natural source of infection.

L223 ....animal is under increasing stress.

Done

L 279 In MxM authors indicate that “the survival of the nematodes was distinct for each sampling area (centre, middle and external sectors) ......and average of the five replications was calculated for the different exposure times, doses and sectors. However, only the average of the three sectors is presented in Tables 1 and 2. No information on the distribution of the nematodes per sector is provided in the result section. I assume their distribution per sector was not meaningful probably due to the low numbers of nematodes used as inoculum in each circle. Please clarify and comment in the revised version.

As the tested products are volatile compounds and they spread uniformly in the space of the Petri dish the average of five replications was calculated for the different exposure times and doses, independently from the sectors. A sentence has been added in MxM

 L295 Volume of the small plastic tube?

The volume of the small plastic tube is 2 ml. It was used to contain the egg masses and to allow their easier recover after treatments

Figure 6. Indicate in the legend that 1= centre, 2= middle and 3 external

Done

Reviewer 2 Report

The manuscript provides some information of two organic compounds that present potential use for plant parasitic nematode control. Results and methodology are appropriate. Minor correction is recommended.

Author Response

The reviewer suggests to change inoculated to infested at page 10 line 270

Ok, we rephrased the sentence

Reviewer 3 Report

Summary:

This study tested the nematicidal effect of two fungal VOCs (1-Octen-3-ol and 3-Octanone) against the root-knot nematode (Meloidogyne incognita) measuring effects on bothjuveniles (J2) and eggs. Results showed that 1-Octen-3-ol was more toxic and acted more quickly than 3-Octanone and than a known nematicide (metham sodium) in J2 assays, while only 1-Octen-3-ol showed substantial impacts on egg-hatch. To assess possible mechanisms of toxicity, they investigated ROS in the nematodes exposed to these VOCs using the flourescent stain dichloroflourescein diacetate (DCFH-DA). Results showed a more rapid accumulation of ROS in J2 treated with 1-Octen-3-ol. They also observed morphological changes in the J2, observing large vacuoles and damage to the digestive tracts of J2 exposed to these compounds. These results suggest that 1-Octen-3-ol shows promise as a potential nematicide and overall the research is advances research on effects of potential nematode toxins for control of and important plant parasitic nematodes.

Strengths: This is primarily a paper screening the efficacies of these metabolites in killing Meloidegyne J2. However, the strenghts of the research are that it investigates interesting volatile compounds that has been shown to have activity in insects comparied to a majority of previous studies that have tested only compounds in liquid filtrates from microbes.  The most interesting aspect of this study is the staining for ROS, which shows a clear correlation with toxicity data and could prove a useful approach for screening efficacy

Weaknesses: It was unclear why they used the known nematicide (metham sodium) at a much lower same concentration than the other volatiles, it makes the results difficult to compare. It is not surprising that these VOCs are more effective than the metham sodium at the minimum recommended dose. The results would have stronger to have also tested multiple concentrations of this product.

Specific suggestions/points for revision:

1) I was unclear what was the concentration of the two volatiles applied. The amount is listed in uL, but I could not find a mention of what was the concentration of the solution or how much total compound applied. Please clarify in the methods.

2) The morphological results of increased size and number of vacuoles has been observed in other studies of nematicidal compounds. The discussion should probably cite and discuss how these results compare. The effect on the digestive tract may be novel, but all of these morphological changes should be placed in the context of previous studies.

Author Response

Weaknesses: It was unclear why they used the known nematicide (metham sodium) at a much lower same concentration than the other volatiles, it makes the results difficult to compare. It is not surprising that these VOCs are more effective than the metham sodium at the minimum recommended dose. The results would have stronger to have also tested multiple concentrations of this product.

We appreciate the valuable reviewer’s comment. We used the commercial nematicide metham sodium at the minimum rate recommended by EU legislation. We cannot exclude that at higher doses metham sodium could be more effective, although the products have different composition.

Specific suggestions/points for revision:

1) I was unclear what was the concentration of the two volatiles applied. The amount is listed in uL, but I could not find a mention of what was the concentration of the solution or how much total compound applied. Please clarify in the methods.

The products commercialized by Sigma-Aldrich (≥98% purity) have a density of 0.8 g/ml. The calculated concentration is 784 g/L as indicated at L245 in material and methods.

2) The morphological results of increased size and number of vacuoles has been observed in other studies of nematicidal compounds. The discussion should probably cite and discuss how these results compare. The effect on the digestive tract may be novel, but all of these morphological changes should be placed in the context of previous studies.

We added a sentence concerning previous results obtained with other nematicidal VOCs showing morphological changes in nematode internal tissues.